# Netrin-1 as a Multitarget Barrier Stabilizer in the Peripheral Nerve after Injury

**DOI:** 10.3390/ijms221810090

**Published:** 2021-09-18

**Authors:** Jeremy Tsung-Chieh Chen, Lea Schmidt, Christina Schürger, Mohammed K. Hankir, Susanne M. Krug, Heike L. Rittner

**Affiliations:** 1Center for Interdisciplinary Pain Medicine, Department of Anesthesiology, Intensive Care, Emergency Medicine and Pain Therapy, University Hospital of Würzburg, 97080 Würzburg, Germany; chen_j1@ukw.de (J.T.-C.C.); LeaSchmidt.mail@web.de (L.S.); christina.schuerger@icloud.com (C.S.); 2Department of Experimental Surgery, University Hospital of Würzburg, 97080 Würzburg, Germany; hankir_m@ukw.de; 3Clinical Physiology/Nutritional Medicine, Campus Benjamin Franklin, Charité-Universitätsmedizin Berlin, 12203 Berlin, Germany; susanne.m.krug@charite.de

**Keywords:** neuropathic pain, netrin-1, blood-nerve barrier, tight junction proteins

## Abstract

The blood–nerve barrier and myelin barrier normally shield peripheral nerves from potentially harmful insults. They are broken down during nerve injury, which contributes to neuronal damage. Netrin-1 is a neuronal guidance protein with various established functions in the peripheral and central nervous systems; however, its role in regulating barrier integrity and pain processing after nerve injury is poorly understood. Here, we show that chronic constriction injury (CCI) in Wistar rats reduced netrin-1 protein and the netrin-1 receptor neogenin-1 (Neo1) in the sciatic nerve. Replacement of netrin-1 via systemic or local administration of the recombinant protein rescued injury-induced nociceptive hypersensitivity. This was prevented by siRNA-mediated knockdown of Neo1 in the sciatic nerve. Mechanistically, netrin-1 restored endothelial and myelin, but not perineural, barrier function as measured by fluorescent dye or fibrinogen penetration. Netrin-1 also reversed the decline in the tight junction proteins claudin-5 and claudin-19 in the sciatic nerve caused by CCI. Our findings emphasize the role of the endothelial and myelin barriers in pain processing after nerve damage and reveal that exogenous netrin-1 restores their function to mitigate CCI-induced hypersensitivity via Neo1. The netrin-1-neogenin-1 signaling pathway may thus represent a multi-target barrier protector for the treatment of neuropathic pain.

## 1. Introduction

Netrin-1 is a secreted chemotropic ligand governing cell migration and precise axon guidance in embryonal development. Netrin-1 is special because its reciprocal actions depend on the receptor of the target cell. It promotes axon path finding and nerve regeneration through its interaction with the canonical receptors deleted in colorectal cancer (Dcc) or the homolog neogenin-1 (Neo1) [1], while it acts as a repulsion molecule through the receptors uncoordinated (Unc5b) and Dcc/Unc in nerve regeneration [2]. The transmembrane netrin receptors belong to the immunoglobulin superfamily. Apart from being a bi-functional neuronal guidance molecule, netrin-1 is involved, amongst others, in resolution of inflammation, angiogenesis, adhesion atherosclerosis, and cancer. Recently, it has been shown that netrin-1 enhances endothelial barrier properties in the blood brain barrier (BBB) [3] and the blood retinal barrier [4]. Netrin-1 upregulates selected tight junction proteins and thereby strengthens these barriers.

Peripheral nerves are guarded by two barriers: the blood-nerve barrier (BNB) and myelin barrier regulate the ingress and egress of substances through axon fibers to maintain the neural microenvironment and preserve optimal nerve function [5,6,7]. The BNB is comprised of endothelial cells shielding endoneurial capillaries and perineurial cells sealing the perineurium. Tight and adherence junction proteins and cytoplasmic accessory proteins form the cell–cell contacts of the BNB and myelin barrier. Claudins and occludin belong to the group of tight junction proteins; zona occludens-1 is located intracellularly to connect tight junction proteins with the cytoskeleton. The myelin barrier is found in the paranode and the mesaxon of Schwann cells [8]. Breakdown of peripheral nerve barriers and its accompanying tight junction protein downregulation are common features of nerve injury [9]. For example, chronic constriction injury (CCI) of the sciatic nerve results in downregulation of claudin-1, claudin-5, and occluding expressions [10,11,12]. After nerves crush claudin-1 and -19 expression are reduced [13], thereby impairing the BNB and myelin barrier function with increased paracellular influx [14]. These findings suggest that peripheral nerve barriers play an important role in the environment for pain-processing and are possible targets for novel analgesic drugs.

The myelin barrier in Schwann cells is comprised of several well-characterized barrier proteins, including occludin, tricellulin, zonula occludin-1, claudin-12, and claudin-19 [5,6]. They cooperate with the paracellular transport proteins to regulate the influx of various ions and molecules [8]. Disruption of the myelin sheath in Jamc KO mice is associated with neuropathic pain [15]. After nerve trauma, peripheral axons degenerate, but Schwann cells dedifferentiate to support axon path finding; however, the myelin barrier has not been studied so far. Nerve injury incites Schwann cell proliferation, migration, and remyelination [16,17] through the release of netrin-1 and activation of the Unc5b signaling cascade [18,19,20,21]. Thus, we hypothesized that netrin-1 also regulates myelin barrier integrity, thereby ameliorating neuropathic pain.

In this study, we analyzed the mechanism by which netrin-1 rescues nerve injury-induced downregulation of tight junction proteins, leaky barriers, and hypersensitivity via a certain netrin-1 receptor, neogenin-1.

## 2. Results

### 2.1. Netrin-1 Dysregulation in the Sciatic Nerve after CCI

The role of netrin-1 was investigated in rats for up to three weeks after CCI surgery. Compared to sham rats, ipsilateral limbs were more sensitive to mechanical stimuli (manual von Frey filaments test (mVF)) and had lower thresholds for thermal sensation (Hargreaves test) over the whole period of three weeks (Figure 1A,B). Hypersensitivity was associated with up to 2.6-fold increased levels of *Ntn1* in the sciatic nerve, starting 6 h after injury and persisting until 21 d (Figure 1C). Since the full neuropathic phenotype was developed at 7 d, subsequent analyses focused on this time point. In contrast to mRNA expression, netrin-1 immunoreactivity in the sciatic nerve was reduced by 58% (Figure 1E). On the receptor side, *Neo1* was much more highly expressed in the sciatic nerve than *Unc5b* and *Unc5c* (Figure 1D). *Neo1* was downregulated 7 d after injury. In conclusion, in CCI, netrin-1 protein and its receptor, *Neo1*, are insufficiently expressed.

### 2.2. Exogenous Recombinant Netrin-1 Treatment Rescues Neuropathic Pain via Neo1

Due to the lack of netrin-1 protein, we hypothesized that exogenous netrin-1 might improve neuropathic pain and neuropathy. Daily application of recombinant netrin-1 (10 µg/day i.p.) gradually reduced the CCI-induced mechanical hypersensitivity to von Frey filament stimulations and thermal hypersensitivity in the Hargreaves test (Figure 2A,B). This treatment also normalized the 70 kDa netrin-1 protein expression in the sciatic nerve. No netrin-1 degradation (i.e., increased formation of 55 kDa netrin fragments) was observed after CCI (Figure 2C–E). Netrin-1 treatment also suppressed CCI-induced upregulation of *Ntn1* in the sciatic nerve (Figure 2F), but *Neo1* expression was unaffected (Figure 2G). Because of netrin-1′s anti-inflammatory properties, we examined the expression of TNFα as an exemplary cytokine. The CCI-induced *Tnfα* upregulation was partially, but not completely, suppressed by netrin-1 treatment in the sciatic nerve (Figure 2H). Overall, systemic netrin-1 administration relieved neuropathic pain and normalized nertin-1 in the sciatic nerve.

We then asked whether exogenous netrin-1 was acting locally: daily peri-sciatic netrin-1 injection also reversed the CCI-induced mechanical and thermal hypersensitivity in a similar time frame (Figure 3A,B). At the low concentration (0.5 µg or 1 µg), the netrin-1 effect was slight and variable (Appendix A). Next, we studied possible netrin-1 receptors involved: local *Neo1* suppression by siRNA prevented netrin-1-induced antinociception if netrin-1 was applied systemically. This treatment further increased the *Ntn1* (Figure 3C–E). Reduction in *Neo1* confirmed effective ablation by *Neo1*-siRNA treatment (Figure 3F).

### 2.3. Netrin-1 as A Multitarget Barrier Stabilizer in the Sciatic Nerve

We then examined the mechanism on netrin-1 action. No changes in tight junction proteins were observed in the dorsal root ganglion (Appendix A), meaning netrin-1′s systemic effects were excluded. Thus, we explored whether netrin-1 could reconstruct the blood–nerve barrier when relieving neuropathic pain. *Cldn1, Cldn5,* and *Cldn19* are typical tight junction proteins of the perineurial, the capillary, and myelin barrier, respectively. They were downregulated by 53–70% one week after CCI injury. Systemic netrin-1 injection effectively rescued *Cldn1*, *Cldn5,* and *Cldn19* expression and the corresponding proteins (Figure 4A,B,D and Figure 5A,B,D).

The claudin-5 and claudin-19 proteins were completely restored, while claudin-1 increased significantly but not completely. This was accompanied in part by a functional recovery: the 68 kDa tracer Evans blue albumin leaked 15 times more into the endoneurium after CCI when applied ex vivo at the sciatic nerve, as indicated by the bright red fluorescence in the endoneurium (Figure 4C). However, netrin-1 only slightly but not significantly reverted the perineurial barrier breakdown.

Fibrinogen as a plasma protein is usually not found in the endoneurium. After CCI injury, fibrinogen immunoreactivity increased by 2.58-times, reflecting disruption of the capillary barrier (Figure 4F). Capillary leakage was completely reversed by systemic netrin-1 treatment. Ex vivo immersion of the desheathed sciatic nerve with 70 kDa FITC-dextran was employed to measure myelin barrier permeability. A total of 70 kDa FITC-dextran significantly accumulated (7.54-fold more) inside of teased nerve fibers after CCI, indicating a leaky myelin barrier (Figure 5C). This was completely reversed by netrin-1 treatment. In parallel, the reduced claudin-19 immunoreactivity in the paranodes after CCI was restored by systemic injection of netrin-1 (Figure 5D).

## 3. Discussion

The neuronal guidance molecule netrin-1 is not only important in neuronal development in embryogenesis, axon guidance, and Schwann cell proliferation, but it apparently also rescues nociceptive hypersensitivity and nerve barrier function after nerve trauma (Figure 6). Local netrin-1 signals via neogenin-1 to target and seal the endoneurial and myelin barrier but not the perineurial barrier. Our data support netrin-1 as a beneficial multitarget barrier-sealing molecule after nerve injury.

Traumatic nerve injury is characterized by barrier breakdown and tight junction protein downregulation [9,10,11,22]: claudin-1, the major protein in the epi-perineurium, regulates the perineurial fence and the spinal cord barrier integrity [23,24]. Claudin-5 is mainly in the endothelial layer of blood vessels of the BNB and the BBB [25]. Both have been shown to be downregulated after CCI [10,11,12]. Claudin-1 is also affected in a later phase in diabetic polyneuropathy [26]. Long-term downregulation or intracellular redistribution of claudin-1 are responsible for perineurial leakage for large and small molecules [12,26]. A decrease in claudin-5 allows for capillary leakage in neuropathy [10,11]. Thus, our results of a loss of certain tight junction proteins are in line with previous data.

Claudin-19 is critical for the paranodal myelin tightness: deficient mice suffer from motor behavior dysfunction attributed to a defect in the paranode and a leaky myelin barrier in the sciatic nerve [27]. These mice also have an abnormal nerve signal propagation of large fibers [28]. The role of the myelin barrier in pain has not been examined before. Here, we show that claudin-19 is reduced in the paranode, which allows for large molecule penetration. We propose that leakiness of the myelin barrier, e.g., slightly myelinated Aδ fibers, Aβ fibers, or Remark Schwann cells, could contribute to increased excitability of the nerve fibers. Leaky barriers could allow for inflammatory mediators like protons or cytokines to diffuse close to axons, facilitating nociceptive hypersensitivity. Receptors like transient receptor potential ankyrin 1 (TRPA1) or vanilloid 1 (TRPV1) could be more accessible, mediating lowered post-injury nociceptive thresholds [29,30]. In addition, slower conduction velocities caused by neuropathy could contribute to neuropathic pain [31,32]. Thus, several mechanisms could work together to elicit hyperalgesia after BNB and myelin barrier opening.

Our study supports the role of netrin-1 as a universal barrier stabilizer. This extends findings from previous studies: netrin-1 seals endothelial barriers like the blood–retina barrier in diabetic retinopathy [4] as well as the BBB in stroke [33,34], hemorrhage [35], and experimental autoimmune encephalomyelitis [3]. However, the effects on barriers are more complex. In the retina, structures and tight junction protein expressions are controlled by netrin-1: full 70 kDa netrin-1 seals the barrier, but the 55 kDa netrin-1 VI-V fragment [4], formed by metalloproteinase 9 (MMP9) cleavage, promotes leakage. The VI-V fragment leads intracellularly to the activation of FAK and SRC kinases [4], whereas the signaling for protection is less clear: PI3K and FAK have been implicated [33,35]. Thus, full netrin-1 seals endothelial barriers; however, further intracellular mechanisms need to be evaluated. After nerve injury, we did not find any evidence of netrin cleavage and netrin-1 fragments, suggesting barrier-protective pathways to be of more importance.

The role of netrin-1 in epithelial barriers is less clear. Although netrin-1 treatment is beneficial in colitis, it does not directly influence permeability of the gut, protected by epithelial cells [36]. This would be consistent with our data, because claudin-1 was not completely restored and the function of perineurial barrier was not significantly improved. Thus, netrin-1 seems to be more critical for endothelial barrier resealing.

In our study, we not only found that netrin-1 seals the capillary barrier as part of the BNB but also the myelin barrier, supporting the role as a multitarget barrier stabilizer. Netrin-1 and its receptors, neogenin-1 and Unc5b, are expressed in Schwann cells. Netrin-1 promotes peripheral nerve regeneration and Schwann cell proliferation after nerve crush [37,38]. Neural regeneration occurs via the netrin-1 signaling pathway [38]. However, the time course in early CCI makes effects on regeneration less likely. Regeneration in CCI occurs between 9–12 weeks after injury, making an early effect after one week less plausible and supporting its role in the myelin barrier sealing and thereby antinociception. Secondly, netrin-1 is anti-inflammatory; thus, reduced cytokine levels like *Tnfa* could contribute as well. *Tnfa* elicits hyperalgesia and weakens barriers; thus, *Tnfa* downregulation would induce a double effect. In summary, a multi target netrin-1 downstream network rescues the BNB, the myelin barrier, and neuropathic pain.

We found a discrepancy between the netrin-1 protein and *Ntn1* after CCI. Proteins and mRNA do not always correspond; this, for example, can also be seen for *Cldn2* in intestinal ischemia [39] or *NTN1* in keratinocytes and immunoreactivity in skin samples from patients with small fiber neuropathy [40]. Amounts of proteins and their mRNAs can have a skewed distribution, meaning gene expression sometimes poorly correlates with their protein levels [41]. Various processes, including post-transcription, translation, and degradation, regulate and influence the process of conversion from mRNA to protein in the mammalian cell [42]. In our study, we excluded netrin-1 degradation by, for example, MMP9, because we did not see an increase in the 55 kDa VI-V netrin-1 fragment. In fact, in early CCI, only low levels of MMP9 protein with proteolytic activity are observed in the peripheral nerve [43]. On the level of translation, certain miRNAs, e.g., let-7 or miR-9, could induce *Ntn1* mRNA degradation [44]. Other translational mechanisms could include feedback loops; we postulate that low netrin-1 protein levels might activate its own promotor to induce the transcription of the *Ntn1* sequence.

We found that systemic netrin-1 administration elicited an antinociceptive effect in CCI rats as well as local application, albeit to a lower extent. Since netrin-1 receptors are also expressed in the sciatic nerve, we propose a local effect. Several in vitro studies have shown that netrin-1 protects barrier function and reduces cellular damage [45,46]. A similar protective effect of netrin-1 through systemic or local treatment has been demonstrated in vivo [3,35], suggesting that high concentrations of netrin-1 maintain normal physiological conditions by maintaining barrier sealing. Local application at the nerve is challenging, because proteins easily diffuse away if the space is not confined like in the joint or the spinal canal. Our previous studies have shown that the opening of the nerve barrier leads to nociceptive hypersensitivity responses when inflammation is present [22]. Thus, the netrin-1-induced anti-hypersensitivity effect may depend on its local function related to netrin-1 receptor expression rather than systemic regulation.

What are possible clinical implications? Firstly, if the BNB and myelin barrier are shielding against pain, patients with congenital mutations in tight junction protein genes could also suffer from pain. *CLDN19* mutations cause recessively inherited familial hypomagnesemia with hypercalciuria and nephrocalcinosis (FHHNC). Interestingly, these patients have an abnormal motor function and increased abdominal pain and pain upon exercise [47]. This supports our findings that loss of claudin-19 expression could cause hyperalgesia. On the other hand, *CLDN1* mutations cause neonatal ichthyosis sclerosing cholangitis hypotrichosis (NISCH) syndrome, but pain is not described in these patients [48]. Congenital *CLDN5* mutations have not been described yet. Thus, these rare diseases could help to understand the role of the BNB and the myelin barrier. Secondly, netrin-1 could be used to treat neuropathic pain. Alternatively, small molecules could be designed to activate neogenin-1, although nothing is known to be in the pharmaceutical pipelines yet. Thirdly, netrin-1 and its receptors could be important apart from their barrier stabilizing function. In skin samples of patients with polyneuropathy, *NTN1* in proximal and *DCC* and *NEO1* (two attracting receptors) in distal specimens are reduced [49]. In contrast, increased netrin-1 immunoreactivity from keratinocytes from small fiber neuropathy patients is supposed to promote fiber degeneration in the skin [40].

In summary, netrin-1 is a local, multitarget barrier sealer relieving neuropathic pain via neogenin-1, leading to principal reconstitution of the capillary and the myelin barrier. Further studies should elucidate the intracellular mechanisms to discover possible targets for small molecules enhancing the netrin-1-neogenin-1 endogenous pathway, which could promote pain resolution. Secondly, leakage of the myelin barrier and its resealing are interesting new pathways to target hyperexcitability in neuropathy. Thirdly, the current research results showing the netrin-1 short-term effect may support future research of its effect in the chronic phase of neuropathic pain.

## 4. Materials and Methods

### 4.1. Chronic Constriction Injury (CCI) and Nociceptive Tests

All animal experiments were performed according to the German Animal Protection Law and approved by the local ethics committee (55.2.2-2532-2-612, Regierung von Unterfranken, Würzburg, 06.04.2018). We housed rats in groups of six in cages in light and temperature-controlled rooms under specific pathogen-free conditions (12 h:12 h light/dark cycle, 21–25 °C, 45–55% humidity). Wistar rats (200–300 g) had free access to food and water and were randomized to the treatment vs. control. Under deep anesthesia (2% isoflurane), we placed 4 loose silk ligatures (4/0) (with 1 mm spacing) around the sciatic nerve at the level of the right mid-thigh in rats [22]. Ligatures were tied until they elicited a brief twitch in the respective hind limb. The muscle layer and the incision in the shaved skin layer were closed with suturing material and metal clips separately after the surgery. Rats were monitored and maintained at a controlled temperature (37 °C) until fully recovered from anesthesia.

### 4.2. Nociceptive Behavioral Assays

#### 4.2.1. Thermal Nociceptive Behavioral Responses

The Hargreaves test was used to assess the latency of the thermal nociceptive responses. The sensitivity responses of each limb were averaged from three trials, with a recovery period of 10 min between each test. The cut-off latency was 20 s to avoid tissue damage [22].

#### 4.2.2. Static Mechanical Hypersensitivity

A series of von Frey filaments (North cost medical Inc., Morgan Hill, CA, USA) were assessed to record the withdrawal threshold of the hind paw and identify the allodynia response and touch sensitivity after surgery or drug injection. In general, the filaments were applied to the plantar surface of the hind paw and held for 1–3 s until the filaments were bent to an angle of 45°. We used up–down methods to determine 50% paw withdrawal to von Frey filaments [50].

All rats were sacrificed by CO_2_ asphyxiation after the behavioral test; ipsilateral and contralateral sciatic nerves and skins were harvested for the biochemical and molecular experiments by removing connective tissue and ligatures. All experiments and treatments were performed by a blinded examiner.

### 4.3. Drugs and siRNA Treatment

Rats received recombinant human netrin-1 daily by intraperitoneal injections (10 µg/rat, R&D systems, 6419-N1; Minneapolis, MN, USA), according to a previous mice study and further converted to a suitable dosage for rats by body weight [3]. Rats also received netrin-1 perineurially (0.5, 1, or 2 µg/rat) for 7 d after CCI surgery. Under 2% isoflurane anesthesia, a sharp needle was used to penetrate the skin, and it marked the location of the nerve according to anatomy. The right sciatic nerve was positioned using a 22 G analgesic needle attached to a nerve stimulator (Stimuplex Dig RC; Braun, Melsungen, Germany) to avoid intraneural injection, as described previously [22]. Perisciatic location was verified by sustained leg twitching under a reduced current. Rats received 100 µL of 0.9% NaCl or 2 µg netrin-1 in 100 µL of 0.9% NaCl. Drug treatment animals were sacrificed after the experiment to analyze tissues at that time point. siRNA was delivered via perineurial injection. We used commercial siRNA to suppress local mRNA expression. *Neo1*-targeting siRNA (Silencer^®^ Pre-designed siRNA, siRNA cat#: s135612, Thermo Fisher Scientific, Darmstadt, Germany) and Negative-control siRNA (Silencer^®^Cy3 Labeled siRNA, siRNA cat#: AM4621, Thermo Fisher Scientific, Darmstadt, Germany) or scrambled-siRNA-Cy3 (Mission^®^siRNA universal negative controls, ProductNo. SIC003, Sigma, Taufkirchen, Germany) alone were applied on the surface of the sciatic nerve after surgical preparation, as outlined above. For siRNA delivery, 2 µg of the target siRNA or scrambled-siRNA was mixed with i-Fect™ transfection reagents (Neuromics, Edina, MN, USA) in a ratio of 1:5 (W:V) to a final concentration of 400 mg/L. 

### 4.4. Nerve Permeability

#### 4.4.1. Permeability of the BNB (Perineurium)

To analyze the permeability of the perineurium, we used Evans blue albumin (EBA) [22,24]. EBA was prepared in 5% BSA with 1% EB dye (Sigma-Aldrich, St. Louis, MO, USA) in sterile distilled PBS and filtered through a 0.20 µm filter (Sartorius stedim Biotech GmbH, Göttingen, Germany). Dissected sciatic nerves (2 cm length, proximal to the sciatic trifurcation) from rats were immersed ex vivo in 2 mL of EBA for 1 h and then fixed with 4% paraformaldehyde for 1 h. Afterward, tissues were embedded in Tissue–Tek, cut into 10 µm thick sections, and mounted on microscope cover glasses with permaFluor Mountant (Thermo Scientific, Fremont, CA, USA). Three sections/rat were analyzed, and a mean was calculated. The diffusion of EBA determined the permeability of the perineurium into the endoneurium.

#### 4.4.2. Permeability of the Myelin Barrier

The whole sciatic nerve was desheathed from the epiperineurium. Desheathed sciatic nerves were sealed with vaseline in both ends and incubated in artificial cerebrospinal fluid (as following (mM): 10 HEPES, 17.8 NaCl, 2 NaHCO_3_, 4 MgSO_4_, 3.9 KCl, 3 KH_2_PO_4_, 1.2 CaCl_2_, 10 Dextrose; pH 7.4), containing fluorescein isothiocyanate-dextran (FITC-dextran; 5 mg/mL; MW: 68 kDa; Sigma, Taufkirchen, Germany) for 1 h at 37 °C [28]. Afterwards, nerves were fixed in 4% paraformaldehyde for 5 min at RT, then placed on glass slides and teased into individual fibers under a dissecting microscope. We then mounted these teased fibers by VECTASHIELD Antifade Mounting Medium and covered them with microscope coverslips. The green fluorescence signal was determined.

### 4.5. RNA Extraction and Quantitative PCR

TRIzol^®^ reagent (Invitrogen, Thermo Fisher Scientific, Darmstadt, Germany) was used to extract the RNA of the ipsilateral sciatic nerve tissue. Total RNA (1 µg) was reverse transcribed to cDNA by using a high-capacity cDNA reverse transcription Kit (Applied Biosystems, Thermo Fisher Scientific, Darmstadt, Germany). The primers for RT-qPCR were designed by the Primer3 online data bank. We used the PowerUpTM SYBR^®^ Green Master Mix (Applied Biosystems, Thermo Fisher Scientific, Darmstadt, Germany) detection system to amplify and subsequently measure the PCR product in a 7300 Real-Time PCR System (Applied Biosystems, Thermo Fisher Scientific, Darmstadt, Germany) with the following program: 95 °C for 10 min and 40 cycles at 95 °C for 3 s and 60 °C for 30 s. *Gapdh* served as an endogenous reference gene for quantification [24]. Primer sequences are given in Table 1. Relative mRNA abundances to reference gene mRNA were calculated by the ΔCt method (the threshold cycle value). The comparative cycle threshold (ΔΔC_T_) method was performed to assess mRNA levels of samples from different groups. Final mRNA expression was expressed as 2^-ΔΔCT^), standardized to a value from the control group as depicted in the figure legends.

### 4.6. Western Blotting

The lower red phase of TRIzol^®^ lysate was taken and mixed with 700 µL of isopropanol. The insoluble pellet was obtained by centrifugation at 12,000× *g* for 10 min. The pellet was washed 3 times by 0.3 M guanidine hydrochloride in 95% ethanol for 30 min. The pellet was washed with 100% of EtOH for 20 min and centrifuged at 7500× *g* for 5 min at 20 °C. Extracted ipsilateral sciatic nerve proteins were diluted in a lysis buffer and incubated with a BCA protein assay reagent. Proteins were detected using the specific antibodies rabbit polyclonal anti-claudin-1 (#51-9000, 1:1000, Life Technologies, Darmstadt, Germany), mouse monoclonal anti-claudin-5 (#352500, Thermo Fisher Scientific; 1:1000, Darmstadt, Germany), mouse monoclonal anti-claudin-19 (#SC-365690, Santa Cruz; 1:1000, Dallas, TX, USA), rat monoclonal anti-netrin-1 (#MAB1109, R&D system; 1:1000, Minneapolis, MN, USA), mouse monoclonal anti-claudin-19 (#SC-365690, Santa Cruz; 1:1000, Dallas, TX, USA), and as a protein loading control, β-actin (#A3854, Sigma Aldrich; 1:20,000, Taufkirchen, Germany). Peroxidase conjugated goat anti-rabbit IgG, goat anti-mouse IgG, and the chemiluminescence detection system Lumi-Light PLUS Western blotting kit (Roche, Mannheim, Germany) were used to detect bound antibodies. Quantification was done by densitometry (Alpha Innotech, Santa Clara, CA, USA; FluorChem FC2 Imaging systems, Multilmage II).

### 4.7. Immunofluorescence

The harvested tissues were isolated and embedded in Tissue–Tek (Sakura, Alphen aan den Rijn, The Netherlands). The blocks were cryo-sectioned at 10 µm thickness and mounted on SuperFrost-plus microscope slides and stored at −20 °C. The tissue sections were fixed with 100% ethanol at −20 °C for 20 min and were permeabilized for 7 min with 0.5% Triton X-100 (Sigma-Aldrich, St Louis, MO, USA) in PBS, blocked for 30 min in 10% donkey serum (Sigma, Taufkirchen, Germany) in PBS, and incubated overnight at 4 °C with rat monoclonal anti-netrin-1 (#MAB1109, 1:100, R&D systems, Minneapolis, MN, USA), rabbit polyclonal anti-claudin-1 antibody (#51-9000, Thermo Fisher Scientific; 1:100, Darmstadt, Germany), mouse monoclonal anti-claudin-5 (#352500, Thermo Fisher Scientific; 1:100, Darmstadt, Germany), mouse monoclonal anti-claudin-19 (#SC-365690, Santa Cruz; 1:100, Dallas, Texas, USA), rabbit polyclonal anti-pan NaV (#AB5210, S; 1:100, Taufkirchen, Germany), and goat polyclonal anti-fibrinogen (#ABIN458743, antibodies-online; 1:100, Aachen, Germany), respectively. On the second day, the sections were washed in PBS and incubated for 1 h at room temperature with the following secondary antibodies: goat anti-rat IgG Alexa Fluor 594 (1:800; A1107, Thermo Fisher Scientific, Darmstadt, Germany) donkey anti-goat IgG Alexa Fluor 488 (1:800; A11055, Thermo Fisher Scientific, Darmstadt, Germany), donkey anti-mouse IgG Alexa Fluor 555 (1:800; A31570, Thermo Fisher Scientific, Darmstadt, Germany), and donkey anti-rabbit IgG Alexa Fluor 488 (1:800; A21206, Thermo Fisher Scientific, Darmstadt, Germany); then, they were washed again and mounted with VectashieldTM Mounting Medium (Vector Labs, Burlingame, CA, USA). The sections were viewed using an all-in-one fluorescence microscope (BZ-9000, Keyence, Neu-Isenburg, Germany). Images were analyzed using the free software ImageJ (NIH, Bethesda, MD, USA).

### 4.8. Fluorescence Quantification

ImageJ/Fiji was used to analyze our targeting fluorescence signals. To analyze the fluorescence intensity of the netrin-1 staining, the background was reduced in every picture with the rolling ball algorithm, thereby separating the red, green, and blue channels. Positive immunoreactivity was assumed if a signal was above the background (total FL signal). This was quantified in the area of the endoneurium, as outlined by the dotted lines in the figures. We used two sections per animal and calculated the mean for each animal.

To measure FITC-Dextran signals in teased fibers, we scanned and imaged nerve fibers from rats by fluorescence microscopy at 40X magnification. Microscope and camera settings (e.g., light level, exposure, gain, etc.) were identical for all images. The FITC-dextran signals and the total number of individual fibers were measured and counted by ImageJ/Fiji. After background subtraction, we selected single nerve fibers using bright field images, and we placed a 5 µm diameter circle randomly inside the fibers as the region of interest. These ROIs were then transferred to their corresponding green fluorescence images and analyzed for fluorescence intensity within the ROI. With GraphPad, we calculated the mean gray values, area, and integrated density of ROIs.

### 4.9. Statistical Analyses

Statistical analysis was done using Graph Pad version 9 or R-studio version 1.1.453 software. When comparisons were for more than two groups, a Mann–Whitney-U-test or two-way ANOVA were used, as described in the figure legends presented. For pain behavior tests, a two-way RM ANOVA with Bonferroni’s post-hoc test was employed. For qPCR results regarding mRNA expression, a Mann–Whitney-U-test was selected as a non-parametric test to examine differences between the two groups. Differences between groups were considered significant if *p* ˂ 0.05. All data are expressed as mean ± SEM.

## Figures and Tables

**Figure 1 ijms-22-10090-f001:**
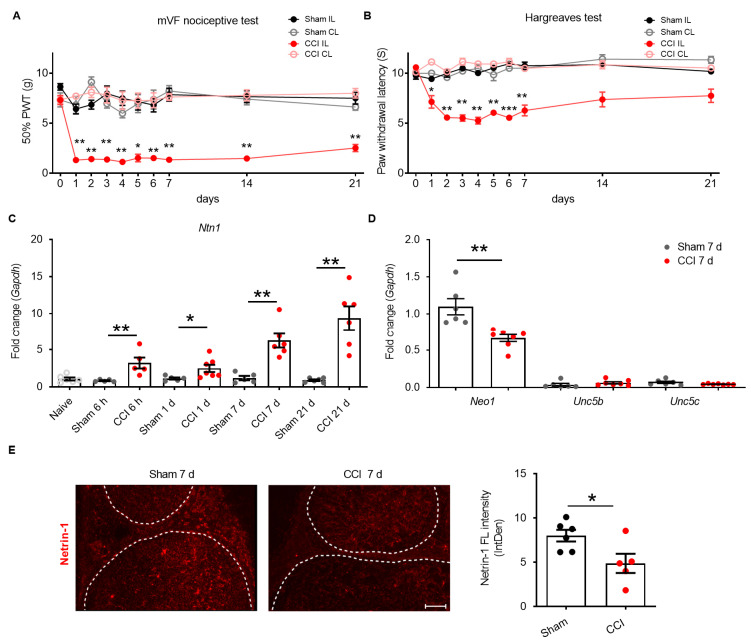
CCI induces hypersensitivity and reduces netrin-1 protein. Wister rats underwent CCI surgery and were analyzed for up to three weeks; sham rats had surgery without nerve ligation. (**A**) Nociceptive thresholds: static mechanical hypersensitivity was detected by von Frey filaments test and (**B**) thermal hypersensitivity in the Hargreaves test (*n* = 10; PWT: paw withdrawal threshold; IL: ipsilateral, CL: contralateral, two-way RM-ANOVA, Bonferroni’s post hoc test). (**C**) *Ntn1* expression in the sciatic nerve of CCI and sham rats at different time points after injury (Mann–Whitney U test). Values were normalized to naïve animals. (**D**) Netrin-1 receptors, *Neo1*, *Unc5b,* and *Unc5c,* expression was analyzed in the sciatic nerve (Mann–Whitney U test; Scale bars: 100 μm). (**E**) Netrin-1-immunoreactivity was analyzed in cross sections of the sciatic nerve. The dotted line delineates the peri-endoneurial border (*n* = 6; FL: fluorescence; IntDen: integrated density). All data are shown as mean ± SEM, * *p* < 0.05, ** *p* < 0.01, *** *p* < 0.001; *n* = 5–6/group.

**Figure 2 ijms-22-10090-f002:**
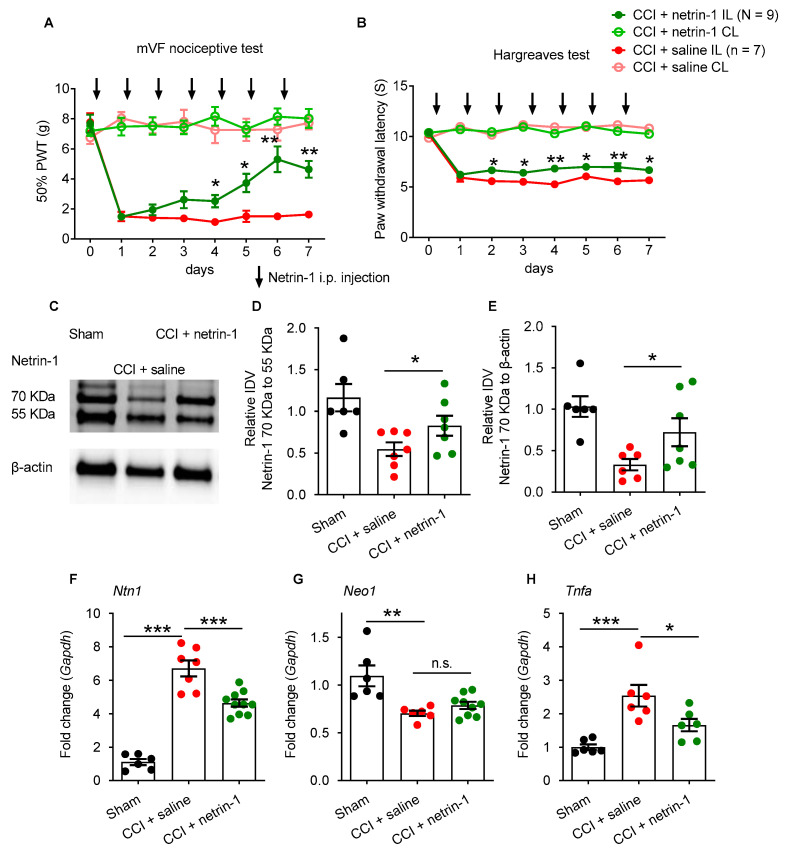
Systemic netrin-1 injection improves injury-induced hypersensitivity and restores netrin-1. Wistar rats with CCI injury were treated with recombinant human netrin-1 injections starting on the day of injury (i.p., 10 µg/rat). Controls rats were treated with a solvent or underwent sham surgery. (**A**) Mechanical nociceptive thresholds (assayed by von Frey filaments) and (**B**) thermal nociceptive thresholds (assayed by the Hargreaves’ test) were measured over 7 d post-injury (two-way ANOVA with Bonferroni’s post hoc test comparing netrin-1 vs. saline, *n* = 7 or *n* = 9/group). The arrows indicate i.p. injection time points. (**C**–**E**) The fragment (55 kDa) and full length (70 kDa) bands of netrin-1 in the sciatic nerve were analyzed by Western blot and quantified by densitometry (IDV: integrated density value; CCI + saline versus CCI + netrin-1, Mann–Whitney U test). (**F**–**H**) *Ntn1, Neo1,* and *Tnfa* expression in the sciatic nerve (Mann–Whitney U test). (C-I) *n* = 6/group. All values were normalized to sham, mean ± SEM, * *p* < 0.05, ** *p* < 0.01, *** *p* < 0.001, n.s.: not significant.

**Figure 3 ijms-22-10090-f003:**
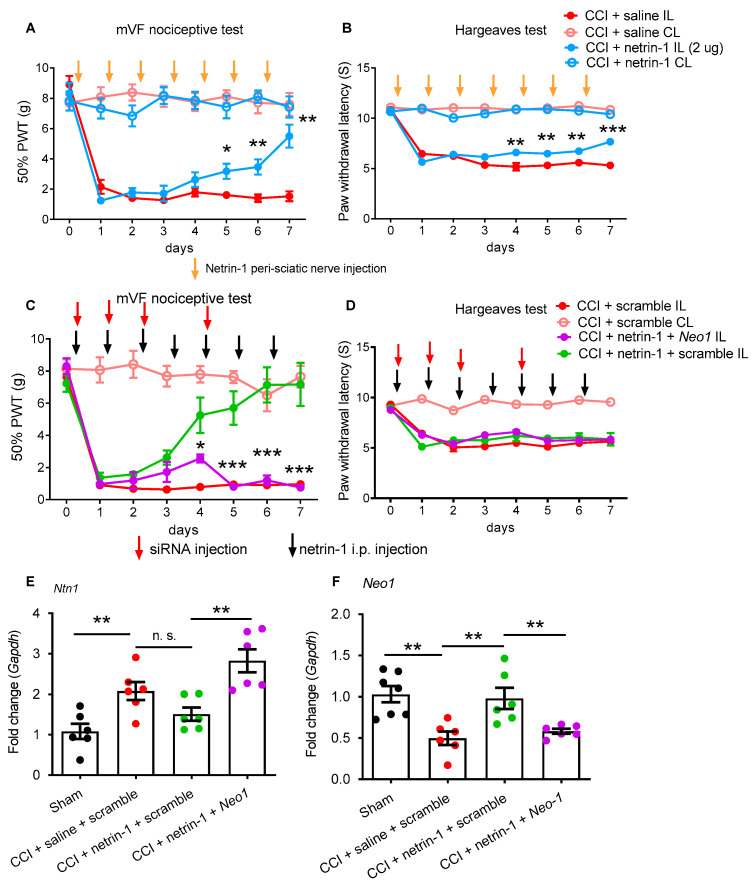
Netrin-1-induced antinociception requires *Neo1* in the sciatic nerve. Wistar rats underwent CCI surgery. (**A**) The time-course analyses of daily peri-sciatic injection of 2 µg netrin-1 was analyzed for the sensitivity to mechanical and (**B**) thermal stimulation (two-way ANOVA with Bonferroni’s post hoc test comparing netrin-1 vs. saline). The orange arrows indicate peri-sciatic injection time points. (**C**,**D**) Rats were treated 4× with perisciatic *Neo1*-siRNA or scramble siRNA (2 µg; red arrows) and daily systemic netrin-1 injections (10µg/rat; black arrows) and tested for mechanical and thermal hypersensitivity (two-way ANOVA with Bonferroni’s post hoc test comparing Neo1-siRNA vs. scramble) as well as *Ntn1* and *Neo1* (**E**,**F**) in the sciatic nerve (Mann–Whitney U test). All mean ± SEM, * *p* < 0.05, ** *p* < 0.01, *** *p* < 0.001, n.s.: not significant *n* = 6.

**Figure 4 ijms-22-10090-f004:**
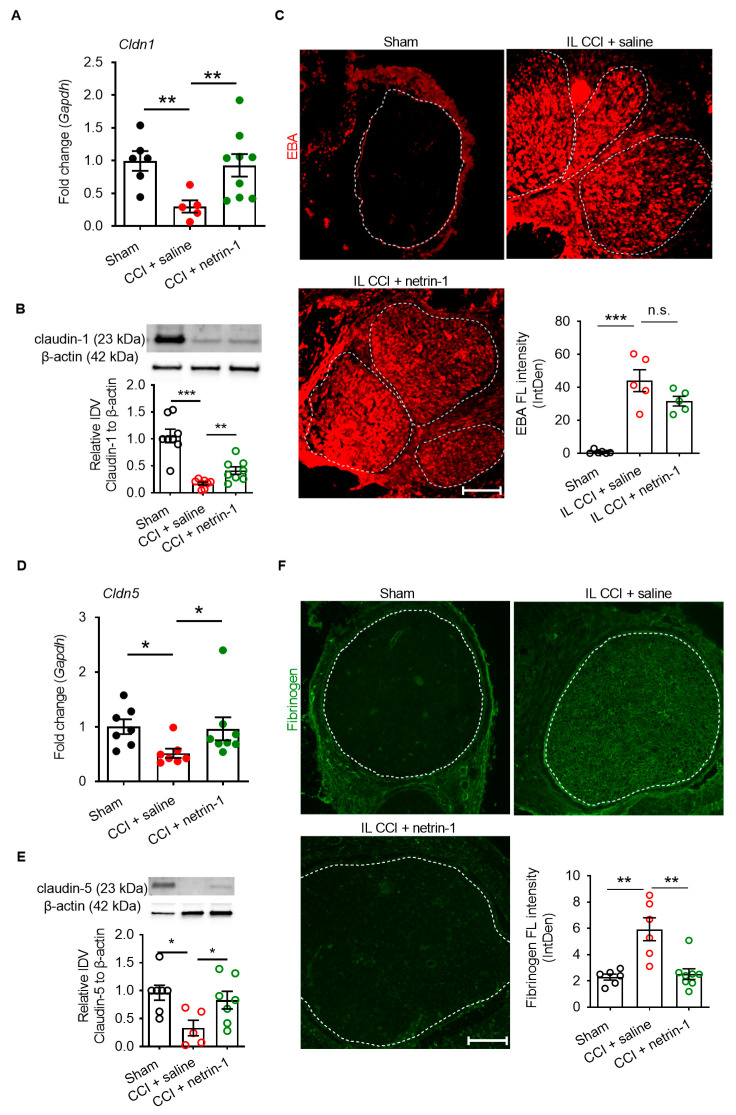
Exogenous netrin-1 stabilizes the BNB after CCI. Wister rats underwent CCI and were treated with daily i.p. netrin-1 injection. (**A**,**B**,**D**,**E**) Tight junction protein mRNA and protein expression of *Cldn1* and *Cldn5* sciatic nerves were quantified (IDV: integrated density value). Gapdh was used as a housekeeping gene for normalization of gene expression experiments, and β-actin for protein analysis experiments. (**C**) Evans blue albumin dye signal (red) was measured after ex vivo immersion of sciatic nerves and subsequently quantified in the endoneurium. If more than one fascicle was present, the mean was calculated (FL: fluorescence; IntDen: integrated density). (**F**) Endoneurial fibrinogen immunoreactivity (green) was analyzed in control and injured sciatic nerves (*n* = 5–7). Dotted lines delineate the endoneurium and area of quantification. All mean ± SEM, *n* = 5–7, Mann–Whitney U test, * *p* < 0.05, ** *p* < 0.01, *** *p* < 0.001, n.s.: not significant. Scale bars: 100 μm.

**Figure 5 ijms-22-10090-f005:**
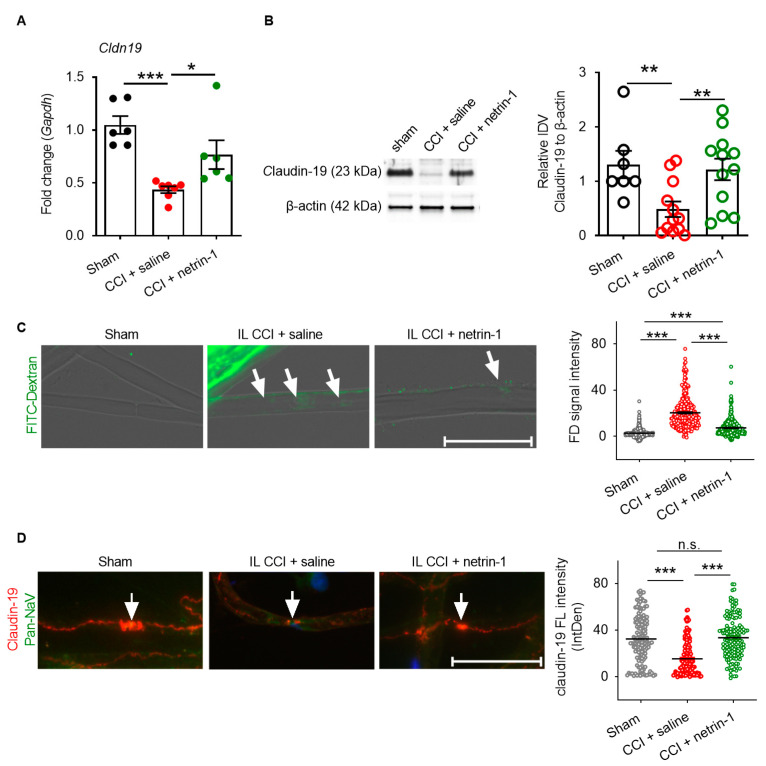
Netrin-1 rescues myelin barrier breakdown. Wistar rats with CCI were treated with daily i.p. netrin-1 injection. (**A**,**B**) *Cldn*19 expression relative to Gapdh and protein relative to β-actin were measured in the sciatic nerve (*n* = 6 and *n* = 7–12, respectively). (**C**) 70 kDa FITC-dextran leakage into teased nerve fiber was analyzed and quantified after ex vivo application of the dye at the sciatic nerve and subsequent fiber teasing (Arrow: FITC-dextran signal, 351–452 fibers from *n* = 5–7 rats/groups; Scale bars: 50 μm). (**D**) Immunoreactivity for claudin-19 (red) and pan-NaV (green) was quantified in teased nerve fibers (Arrow: nodes of Ranvier; 137–161 fibers from *n* = 5–7 rats/group; Scale bars: 50 μm). All mean ± SEM, Mann–Whitney U test, * *p* < 0.05, ** *p* < 0.01, *** *p* < 0.001, n.s.: not significant.

**Figure 6 ijms-22-10090-f006:**
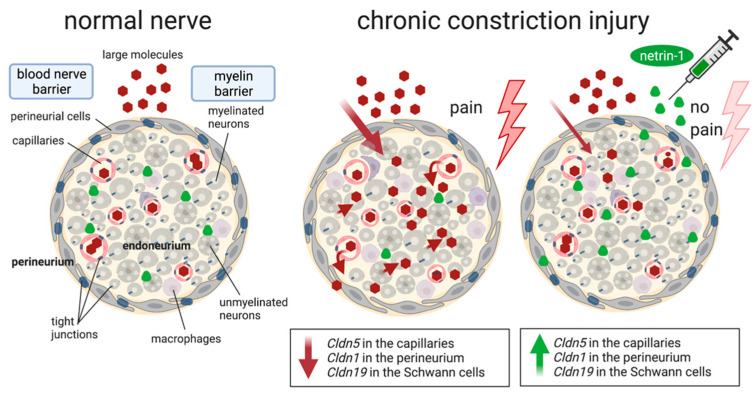
Barrier breakdown in chronic constriction injury and its rescue by the multitarget barrier sealer netrin-1. Nerve ligation results in opening of the myelin barrier and the BNB with corresponding loss of tight junction proteins. Netrin-1 protein remains low. Replacing netrin-1 by exogenous application reverses these changes (especially of the myelin and the capillary barrier) and relieves neuropathic pain. Created with biorender.

**Table 1 ijms-22-10090-t001:** The sequence of primers for RT-qPCR.

Name	Forward	Reverse
*Gapdh*	5′-AGTCTACTGGCGTCTTCAC-3′	5′-TCATATTTCTCGTGGTTCAC-3′
*Cldn1*	5′-GGGACAACATCGTGACTGCT-3′	5′-CCACTAATGTCGCCAGACCTG-3′
*Cldn5*	5′-AAATTCTGGGTCTGGTGCTG-3′	5′-GCCGGTCAAGGTAACAAAGA-3′
*Cldn12*	5′-AACTGGCCAAGTGTCTGGTC-3′	5′-AGACCCCCTGAGCTAGCAAT-3′
*Cldn19*	5′-TGCTGAAGGACCCATCTG-3′	5′-TGTGCTTGCTGTGAGAACTG-3′
*Ntn1*	5′-CAGGAAGGACTATGCTGTCCA-3′	5′-TACGACTTGTGCCCTGCTTG-3′
*Neo1*	5′-TGTGATGGTGACCAAAGGCA-3′	5′-GGAGGCTGCCAGTTCACTATT-3′
*Unc5b*	5′-CGACCCTAAAAGCCGCCCC-3′	5′-GGGATCTTGTCGGCAGAGTCC-3′
*Unc5c*	5′-AGGCTGCTCCTGACTCAGATG-3′	5′-GGGTCTAGAATTGGAGAATTGG-3′

## Data Availability

The data presented in this study are available on request from the corresponding author.

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
