# Peer review of "Netrin-1 as a Multitarget Barrier Stabilizer in the Peripheral Nerve after Injury"

_ijms, 2021, doi:10.3390/ijms221810090_

Round 1

Reviewer 1 Report

Jeremy et al. have shown that activation of netrin-1/Neo1 signal at peripheral nerve could be effective target for the chronic construction injury-induced neuropathic pain. The manuscript ‘netrin-1 signal and pain’ present interesting topic. However, as listed below, there are some points need to be addressed.

Major

  1. Overall: The most critical problem of this manuscript is that control groups are inappropriate. Non-treated sham rats were used for control, to compare the vehicle-, netrin-1- or siRNA-treated CCI rats. This control group do not receive any stress caused by injections. The accuracy of control group could affect the quality of the results. The authors should use appropriate controls for these experiments.
  2. Effect of netrin-1 on the pain threshold should also tested in sham rats. To determine whether the treatment could be therapeutic target or not, checking the effect on sham is important.

Minor

  1. Which side (ipsilateral or contralateral) of sciatic nerves were used for RT-PCR and western blotting?
  2. When assessing the density of immunoreactive signal and FITC-dextran signal, what criteria were used to include a 'positive' cell?
  3. Temperature, humidity and dark/light cycle at the animal room should write to the “Materials and Methods” section.
  4. Please add more detailed description of perineurial injection.
  5. All secondary antibodies used for immunohistochemistry (AlexaFluor-conjugated secondary antibodies) should write to the “Materials and Methods” section.
  6. Although 10 and 2 µg of netrin-1 were injected to i.p. and sciatic nerve, respectively, the rationale for these doses should be added to the manuscript.
  7. As shown in Fig 2, protein expression of full-length netrin-1 in sham group is more abundant than CCI group. Therefore, it is better to add the illustration of netrin-1 to the normal nerve in Fig. 6.
  8. Line 293: One citation for the CCI method seems more than adequate since the way to establish this model is general. Please correct also in the literature list.

Author Response

Response letter to the reviewers

We appreciate the careful revision of the referees and their comments which helped to improve the manuscript. Please find our responses to each referee below. All changes are highlighted in yellow in the revised manuscript.

Reviewer 1

Chen et al. have shown that activation of netrin-1/Neo1 signal at peripheral nerve could be effective target for the chronic construction injury-induced neuropathic pain. The manuscript ‘netrin-1 signal and pain’ present interesting topic. However, as listed below, there are some points need to be addressed.

Major

  1. Overall: The most critical problem of this manuscript is that control groups are inappropriate. Non-treated sham rats were used for control to compare the vehicle-, netrin-1- or siRNA-treated CCI rats. This control group do not receive any stress caused by injections. The accuracy of control group could affect the quality of the results. The authors should use appropriate controls for these experiments.

Answer: Thank you for these valuable comments and questions. In Figure 1, nociceptive thresholds for mechanical and thermal stimuli were recorded in sham-operated control rats. We found that the nociceptive thresholds from experimental rats are similar to those from the contralateral side of CCI rats as well as ipsi- and contralateral sides from sham rats.  Indeed, the analysis results showed no significant differences between the groups. In addition, experiments in recent articles have shown that the nociceptive thresholds measured on the contralateral side of CCI animals are like those on sham-operated animals [1-3]. To follow and support the animal protection principles of the 3Rs, we decided to reduce the use of additional controlled animals in further experiments.

  1. Effect of netrin-1 on the pain threshold should also tested in sham rats. To determine whether the treatment could be therapeutic target or not, checking the effect on sham is important.

Answer: Thank you for this important suggestion. In Figure 2, we showed that treatment with the i. p. netrin-1 injection does not change the nociceptive thresholds of the contralateral limb. We cannot completely exclude that sham surgery leads to alterations, but as pointed out above, this is not likely.

Minor

  1. Which side (ipsilateral or contralateral) of sciatic nerves were used for RT-PCR and western blotting?

Answer: This was corrected in line 385 and 403.

  1. When assessing the density of immunoreactive signal and FITC-dextran signal, what criteria were used to include a 'positive' cell?

Answer: Thank you for this question. We did not count positive cells but rather quantified the immunoreactivity in the selected area. We added details of the quantification in the methods section in lines 438

  1. Temperature, humidity, and dark/light cycle at the animal room should write to the “Materials and Methods” section.

Answer: We added this information in line 314.

Please add more detailed description of perineurial injection.

Answer: This was revised in lines 344.

  1. All secondary antibodies used for immunohistochemistry (AlexaFluor-conjugated secondary antibodies) should write to the “Materials and Methods” section.

Answer: We now added this information in 426.

  1. Although 10 and 2 µg of netrin-1 were injected to i.p. and sciatic nerve, respectively, the rationale for these doses should be added to the manuscript.

Answer: Thank you for this valuable comment. Previous data from intraperitoneal injections in mice showed that the dose of 1 ug netrin-1 per mouse every other day was sufficient to attenuate experimental autoimmune encephalomyelitis (EAE) severity [4]. We used this effective dose and further converted this administered to the rats by body weight. According to Jackson's lab, an 8-week-old mouse is about 20-25g, while we used a rat weighing between 200-300 g. Therefore, we chose 10 ug as our I.P. injection dose. We have changed the sentences in line 344.

We have tested the different doses for perineurial injection. This is now added in the supplementary figure 1. 

  1. As shown in Fig 2, protein expression of full-length netrin-1 in sham group is more abundant than CCI group. Therefore, it is better to add the illustration of netrin-1 to the normal nerve in Fig. 6.

Answer: The graphical abstract was adapted accordingly.

  1. Line 293: One citation for the CCI method seems more than adequate since the way to establish this model is general. Please correct also in the literature list.

Answer: Thank you for this suggestion we have deleted the two references from this part in line 319.

  1. Peng, Z.; Zha, L.; Yang, M.; Li, Y.; Guo, X.; Feng, Z., Effects of ghrelin on pGSK-3beta and beta-catenin expression when protects against neuropathic pain behavior in rats challenged with chronic constriction injury. Sci Rep 2019, 9, (1), 14664.
  2. Guo, S. H.; Lin, J. P.; Huang, L. E.; Yang, Y.; Chen, C. Q.; Li, N. N.; Su, M. Y.; Zhao, X.; Zhu, S. M.; Yao, Y. X., Silencing of spinal Trpv1 attenuates neuropathic pain in rats by inhibiting CAMKII expression and ERK2 phosphorylation. Sci Rep 2019, 9, (1), 2769.
  3. Vacca, V.; Marinelli, S.; Pieroni, L.; Urbani, A.; Luvisetto, S.; Pavone, F., 17beta-estradiol counteracts neuropathic pain: a behavioural, immunohistochemical, and proteomic investigation on sex-related differences in mice. Sci Rep 2016, 6, 18980.
  4. Podjaski, C.; Alvarez, J. I.; Bourbonniere, L.; Larouche, S.; Terouz, S.; Bin, J. M.; Lecuyer, M. A.; Saint-Laurent, O.; Larochelle, C.; Darlington, P. J.; Arbour, N.; Antel, J. P.; Kennedy, T. E.; Prat, A., Netrin 1 regulates blood-brain barrier function and neuroinflammation. Brain 2015, 138, (Pt 6), 1598-612.

Reviewer 2 Report

In this manuscript entitled “Netrin-1 as a multitarget barrier stabilizer in the peripheral 2 nerve after injury”, Jeremy Tsung-Chieh Chen et al. studied the mechanism underlying netrin-1 administration in a preclinical model of neuropathic pain.

I found the manuscript well written and easy to follow. Experimental plan is logical and experiments well executed with appropriate controls.

I just have some minor points:

  • I would revise the abstract. It is not clear what was previously demonstrated (state of art) and what was addressed by the present manuscript.
  • Please better explain why all mechanisms were analyzed up to 7 days instead of following animals up to the chronic phase of the disease. This may be of importance in assessing models of chronic neuropathy. For instance loose of efficacy etc.
  • Fig. 1E netrin-1 staining images are not representative of quantification.
  • It is clear why authors decided to study both i.p. (i.e. systemic effect) and peri-sciatic nerve injection, but I would better highlight this aspect in results and discussion sections.
  • Including data on tight junction proteins in DRG (not shown in the current version of the manuscript) may increase the quality of the manuscript.
  • Please revise abbreviations throughout the manuscript and typos/misspelling (eg netin-1).

Author Response

Response letter to the reviewers

We appreciate the careful revision of the referees and their comments which helped to improve the manuscript. Please find our responses to each referee below. All changes are highlighted in yellow in the revised manuscript.

Reviewer 2

In this manuscript entitled “Netrin-1 as a multitarget barrier stabilizer in the peripheral 2 nerve after injury”, Jeremy Tsung-Chieh Chen et al. studied the mechanism underlying netrin-1 administration in a preclinical model of neuropathic pain.

I found the manuscript well written and easy to follow. Experimental plan is logical, and experiments well executed with appropriate controls.

I just have some minor points:

  • I would revise the abstract. It is not clear what was previously demonstrated (state of art) and what was addressed by the present manuscript.

Answer: Thank you for this valuable suggesting. We have revised the abstract.

  • Please better explain why all mechanisms were analyzed up to 7 days instead of following animals up to the chronic phase of the disease. This may be of importance in assessing models of chronic neuropathy. For instance, loss of efficacy etc.

Answer: Thank you for asking this question. While recording the daily nociceptive thresholds, we analyzed the mean of the nociceptive thresholds for the different groups. We found that the antihyperalgesic effect of netrin-1 on CCI-induced mechanical stimuli was highest after day 6 of daily netrin-1 injection. Since we wanted to analyze this mechanistically first, we did not examine longer treatments. In the future we plan will focus on the effect of netrin-1 on the maintenance phase of CCI. This is now mentioned in the discussion line 307   

  • 1E netrin-1 staining images are not representative of quantification.

Answer: As suggested, we now changed the figures.

  • It is clear why authors decided to study both i.p. (i.e. systemic effect) and peri-sciatic nerve injection, but I would better highlight this aspect in results and discussion sections.

Answer: According to the reviewer’s suggestion, we added a paragraph in the discussion part line 270

  • Including data on tight junction proteins in DRG (not shown in the current version of the manuscript) may increase the quality of the manuscript.

Answers: Thank you for this comment. We have added these results in the supplementary figure 2.

  • Please revise abbreviations throughout the manuscript and typos/misspelling (eg netin-1).

Answers: The manuscript was carefully revised.

Round 2

Reviewer 1 Report

The revised MS has been revised well and is in nice condition now.